# Use of a Single Wireless IMU for the Segmentation and Automatic Analysis of Activities Performed in the 3-m Timed Up & Go Test

**DOI:** 10.3390/s19071647

**Published:** 2019-04-06

**Authors:** Paulina Ortega-Bastidas, Pablo Aqueveque, Britam Gómez, Francisco Saavedra, Roberto Cano-de-la-Cuerda

**Affiliations:** 1Kinesiology Department, Faculty of Medicine, Universidad de Concepción, 4030000 Concepcion, Chile; portegab@udec.cl; 2Electrical Engineering Department, Faculty of Engineering, Universidad de Concepción, 219 Edmundo Larenas St., 4030000 Concepción, Chile; britam.gomez@biomedica.udec.cl (B.G.); frsaaved@gmail.com (F.S.); 3Physiotherapy, Occupational Therapy, Rehabilitation and Physical Medicine Department, Universidad Rey Juan Carlos, 28922 Madrid, Spain; roberto.cano@urjc.es

**Keywords:** timed up & go test, activity segmentation, inertial sensors

## Abstract

Falls represent a major public health problem in the elderly population. The Timed Up & Go test (TU & Go) is the most used tool to measure this risk of falling, which offers a unique parameter in seconds that represents the dynamic balance. However, it is not determined in which activity the subject presents greater difficulties. For this, a feature-based segmentation method using a single wireless Inertial Measurement Unit (IMU) is proposed in order to analyze data of the inertial sensors to provide a complete report on risks of falls. Twenty-five young subjects and 12 older adults were measured to validate the method proposed with an IMU in the back and with video recording. The measurement system showed similar data compared to the conventional test video recorded, with a Pearson correlation coefficient of 0.9884 and a mean error of 0.17 ± 0.13 s for young subjects, as well as a correlation coefficient of 0.9878 and a mean error of 0.2 ± 0.22 s for older adults. Our methodology allows for identifying all the TU & Go sub–tasks with a single IMU automatically providing information about variables such as: duration of sub–tasks, standing and sitting accelerations, rotation velocity of turning, number of steps during walking and turns, and the inclination degrees of the trunk during standing and sitting.

## 1. Introduction

The number of people over 60 years of age is rapidly increasing worldwide. The main reasons for this demographic change are the increase in life expectancy and the fall in the birth rate [1]. This has become a public health problem, since aging is generally associated with a decrease in physical and psychological capacity, as well as an increase in the risk of disability, dependence and a number of comorbidities [1,2].

One of the consequences generated by aging is the increased risk of falls, which have been defined as accidental events in which the person falls after losing control of the center of gravity and no effort is made to restore the balance, or it is inefficient [3].

Falls represent a major public health problem in the elderly population. Approximately one third of the population over 65 years old has experienced at least one fall per year. In addition, this frequency increases by 50% in individuals older than 85 years. Between 20% and 30% of falls result in an injury that requires medical attention, constituting the leading cause of death or non-fatal injury in older adults [4,5,6]. Therefore, the early detection of the decreased function of the elderly is important to initiate precociously the preventive measures that allow for maintaining their functional independence [7].

Normally, rehabilitation professionals perform evaluations through observation, as well as through the application of scales or assessment instruments, which provide a certain level of objectivity to the evaluations. There is a large number of tests and scales that allow for evaluating and assessing the static balance, dynamic balance and gait of healthy subjects or with motor problems [8], the Timed Up & Go test (TU & Go) being the most used worldwide. This test measures the dynamic balance and functional mobility in older adults, as well as in the neurological population [9,10,11].

The TU & Go is a simple test that can be performed anywhere, and consists of the subject getting up from a chair from the sitting to the bipedal position, walking three meters, turn, returning and sitting on the chair again, as illustrated in Figure 1. The variable measured is the total time taken by the test and then the score assigned in seconds is observed, which is correlated with the risk of falls [9,12].

Some of the advantages of the TU & Go test is the simplicity in its application and its short duration. In addition, it requires little equipment and allows the possibility that people with a functional impairment can perform the test. However, one of the limitations is that the TU & Go, although it provides relevant information about the risk of falls, is not capable of determining subjects with greater difficulties objectively. Barry et al. [13] mentioned that a limitation in the predictive value of the test could be explained by the fact that it is a unique test that evaluates the overall balance and mobility, which can be improved by combining it with technological tools for the movement analysis, such as optoelectronic laboratories with passive reflective markers, considered the gold standard instrument for analysis of human movement, or several alternatives, like wireless motion sensors such as inertial sensors or inertial measurement units (IMU).

Optoelectronic laboratories, despite providing accurate measurements, are expensive and their application takes a long time, since training and experience is required to interpret the results. In addition, in several countries, there are rural or remote locations with no resources for these advanced technological evaluations systems.

In recent years, a variety of evidence has been observed regarding the development of different devices that use inertial sensors, applications and/or smartphones as a low-cost alternative to optoelectronic systems, which have allowed for specifically visualizing the phases in which subjects could present greater problems with the consequent probability to fall during the application of the TU & Go test [14,15]. Different authors [7,8,11] indicate that the phases of the TU & Go test correspond to: the transition from sitting position to standing, walking towards the turning mark, turning, walking back to the chair, turning and the sit back. This has been shown to increase the predictive value of the test in relation to the risk of falls and the phases in which subjects present greater difficulties.

Thus, it has been demonstrated that it is possible to implement an automatic segmentation of these phases or activities through feature-based algorithms [11,16], complex algorithms based on machine learning [7,17] or by principal components analysis (PCA) [18]—as techniques based in Wavelet decomposition [19]—using inertial sensors. Feature-based algorithms have the advantage of being simple to implement, but their performance is diminished due to the great variability that exists in the morphology of the signals they use to perform the segmentation—angular velocity and acceleration—those that depend on the environment and the execution time of the activities to be identified. Algorithms based on machine learning have a good response to the great variability of the characteristics present in the signals to be processed, but they are complex in their implementation—like those that use principal components analysis—and, to guarantee the above, they depend on a large amount of data where the mentioned variabilities are presented.

One way to reduce the disadvantages of feature-based methods is using the orientation data from Inertial Measurement Units (IMU) [14,20]. This orientation angles have been used in different parts of the body for the segmentation of activities during the execution of the test, since they have low variability in their inter-subject characteristics, allowing for extracting characteristics of each phase independently [21,22].

In relation to the above, this study presents an automatic segmentation methodology that uses a feature-based algorithm to identify the typical sub-tasks carried out during a three-meter TU & Go test in two groups of subjects (young and older adults) using the orientation angles of a single wireless IMU on the back (L3–L4, approximately) to analyze independently the data of the inertial sensors, in order to provide a complete report on the risk of falls and promote the use of low–cost technological objective elements and simple use in hospitals or rehabilitation centers of rural or remote locations.

## 2. Material and Methods

### 2.1. Design and Setting

A study with a descriptive design is presented, in which experimental tests were performed to analyze the segmentation of the TU & Go test stages, contrasted with the measurements obtained using a wireless IMU on the lower–back. A total of 25 healthy young subjects (18 men and 7 women) between 25 and 33 years old, and 12 elderly subjects (7 men and 5 women) between 59 and 93 years old were recruited in the city of Concepcion, Chile.

Exclusion criteria were the diagnosis of a neurological, vestibular, musculoskeletal or systemic disease that could alter the ability to walk; the diagnosis of any cardiovascular, respiratory or metabolic disease or other conditions that could interfere with the present study; having undergone surgery on the trunk and lower limbs at least two years prior to the present study, the use of assistive devices for walking and the presence of serious visual alterations that could alter the gait pattern.

The measurements were carried out at the Biomedical Engineering laboratory and in the facilities of the Kinesiology department of the Universidad de Concepción. Prior to the measurements, the test was explained to the participants and two trials without data collection were performed to check the understanding of each participant. The authorization was requested through informed consent, which was approved by the Biosecurity, Bioethical and Ethical Committee of the University of Concepción (Number 3180551).

### 2.2. IMU Sensor

A homemade IMU—developed at the laboratory of Biomedical Engineering of the Universidad de Concepción—was used.

The chip sensor utilized [23] has a three-axis accelerometer, a three-axis gyroscope and a three-axis magnetometer (see Table 1), as well as an embedded internal processor able to fuse the magnetic and the inertial data using an extended Kalman filter to accurately deliver the orientation in quaternions, to avoid singularities presents in the Euler and Navigation representations. Then, the orientation data is obtained in angle representation with an accuracy of ±1 deg. The data is sampled at 100 Hz with a low cost 32–bit microcontroller with an Advance RISC (reduce instruction set computer) Machine (ARM) Cortex-M0+ processor and sent to a software application via Bluetooth 3.0 up to a maximum distance of 20 m without risk of occlusion. The entire system is powered by a 500-mAh LiPo battery, which gives 10-h of autonomy.

Figure 2 shows the implemented sensor and its disposition on the subjects.

### 2.3. Test Procedure

The developed sensor is positioned on the back at the height of L3–L4 for young and elderly subjects. It has been shown that the use of a single IMU in that position allows for the detection of all gait events, biomechanical elements of the pelvis and other spatial and temporal kinematics factor [24,25,26].

The TU & Go test was performed following the recommendations of [22]. Three meters away from the chair, a cone was used to mark the location where the patients had to make the turn. Before carrying out the tests, the procedure was explained with a demonstration, to resolve and clarify doubts. Each participant was recorded at 60 fps using a GOPRO HERO 7 high resolution digital video camera (GoPro, Inc., San Mateo, CA, USA) using lineal FOV (Field of View) mode to reduce the image distortion.

Three TU & Go test repetitions were performed by the older adults group, using for the analysis the performance of the higher time of the test carried out. For the young group, only one test was carried out.

Figure 3 shows the set up used for the measurements.

### 2.4. Segmentation Algorithm

For the segmentation of standing, walking, turning and sitting activities, an algorithm was designed. This algorithm processed inclination (Pitch) and rotation (Yaw) signals of an inertial sensor placed on the back of a subject at L3–L4, approximately.

#### 2.4.1. Standing/Sitting Events Identification

To determine the events of standing and sitting, the Pitch signal was used. This corresponds to the inclination actions during the activities to be identified. When a subject is standing or sitting, he tends to make a slight forward inclination with respect to the resting position until it regains its initial inclination (see Figure 4), which can even be observed in subjects with reduced mobility [27].

In order to condition the Pitch signal, this is smoothed by an average filter of order *N* = 5 (see Equation (Equation 1)) and normalized by the absolute maximum of the smooth signal, as seen in Equation (Equation 2):(1)$$Pitch_{smooth}\left(n\right)=\frac{1}{N}\left[Pitch\left(n\right)+Pitch\left(n+1\right)+...+Pitch\left(n+N-1\right)\right],$$
(2)$$Pitch_{norm}\left(n\right)=\frac{Pitch_{smooth}\left(n\right)}{max(|Pitch_{smooth}|)}.$$

Then, the Pitch signal is processed using a local maximum detector, finding $T_{peak1}$ and $T_{peak2}$, that corresponds to the maximum angle of inclination in the standing and sitting events, respectively (see Figure 5).

Then, the difference between samples to the left of $T_{peak1}$ is calculated, as indicated in Equation (Equation 3), and to the right of $T_{peak1}$, as indicated in Equation (Equation 4), to start the search for the standing action, were *i* is a sample iterator. Thus, the start and end of standing action are obtained using Equations (Equation 5) and (Equation 6), respectively, with a factor of 0.05 to determine the threshold slope stop, which was found experimentally:(3)$$\Delta_{1}=\left|Pitch_{norm}\left(Tpeak1-i-0.1\right)-Pitch_{norm}\left(Tpeak1-i\right)\right|<0.05,\;\;\;i=0,0.1,0.2,\ldots ,$$
(4)$$\Delta_{2}=\left|Pitch_{norm}\left(Tpeak1+i+0.1\right)-Pitch_{norm}\left(Tpeak1+i\right)\right|<0.05,\;\;\;i=0,0.1,0.2,\ldots ,$$
(5)$$standing_{i}=T_{peak1}-i,$$
(6)$$standing_{f}=T_{peak1}+i.$$

Similarly, through the same methodology, the sitting action is sought using the difference between samples to the left of $T_{peak2}$, as indicated in Equation (Equation 7), and to the right of $T_{peak2}$, as indicated in Equation (Equation 8). Then, the start and end of standing action are obtained using the Equations (Equation 9) and (Equation 10), respectively:(7)$$\Delta_{3}=\left|Pitch_{norm}\left(Tpeak2-i-0.1\right)-Pitch_{norm}\left(Tpeak2-i\right)\right|<0.05,\;\;\;i=0,0.1,0.2,...,$$
(8)$$\Delta_{4}=\left|Pitch_{norm}\left(Tpeak2+i+0.1\right)-Pitch_{norm}\left(Tpeak2+i\right)\right|<0.05,\;\;\;i=0,0.1,0.2,...,$$
(9)$$sitting_{i}=T_{peak2}-i,$$
(10)$$sitting_{f}=T_{peak2}+i.$$

Figure 6 shows the final result of the search method and Figure 7 shows the proposed standing/sitting activities identification method.

#### 2.4.2. Turning Events Identification

To determine the turns around the 3-m mark and prior to sitting, Yaw is used, which corresponds to changes in the orientation of the sensor during turns. When a subject performs the TU & Go test, it is subjected to a circuit that forces it to make turns of 180 degrees approximately, which can be unequivocally measured by the sensor in the back, since, during turns, the sensor also changes its orientation next to it (see Figure 8).

To condition the orientation change signal, as for the inclination signal, this is smoothed by an average filter of order *N* = 5 (see Equation (Equation 11)) and normalized by the absolute maximum of the signal, as seen in Equation (Equation 12):(11)$$Yaw_{smooth}\left(n\right)=\frac{1}{N}\left[Yaw\left(n\right)+Yaw\left(n+1\right)+...+Yaw\left(n+N-1\right)\right],$$
(12)$$Yaw_{norm}\left(n\right)=\frac{Yaw_{smooth}\left(n\right)}{max(|Yaw_{smooth}|)}.$$

Then, the rotation signal is derived and processed to identify the maximum value max(d/dt) and the minimum value min(d/dt), which are useful to identify the start and end points of the first turn and the second turn, respectively, as seen in Figure 9.

Thus, through a sliding window of 0.1 s or 10 samples (Equations (Equation 13) and (Equation 14)), the start of the 3-m tuning is searched by calculating the mean of the samples in the window to the left of max(d/dt) and the end of the 3-m tuning calculating the mean of the samples in the window to the right of max(d/dt) using Equations (Equation 15) and (Equation 16), respectively: (13)$$W_{1}=\left[T_{max(d/dt)}-10-i,T_{max(d/dt)}-i\right],$$
(14)$$W_{2}=\left[T_{max(d/dt)}+i,T_{max(d/dt)}+10+i\right],$$
(15)$$\bar{W_{1}}>0.02,\;\;\;i=0,0.01,0.02,0.03,...,$$
(16)$$\bar{W_{2}}<0.9,\;\;\;i=0,0.01,0.02,0.03,...$$

Finally, the start and the end of the 3-m turning are obtained through Equations (Equation 17) and (Equation 18), respectively:(17)$$turn\;1_{i}=min\left(W_{1}\right),$$
(18)$$turn\;1_{f}=max\left(W_{2}\right).$$

Then, using a sliding window of 0.1 s or 10 samples (Equations (Equation 19) and (Equation 20)), the start of the pre-sitting tuning is sought by calculating the mean of the samples in the window to the left of min(d/dt) and the end of the pre-sitting tuning calculating the mean of the samples in the window to the right of min(d/dt) using Equations (Equation 21) and (Equation 22), respectively:(19)$$W_{3}=\left[T_{min(d/dt)}-10-i,T_{min(d/dt)}-i\right],$$
(20)$$W_{4}=\left[T_{min(d/dt)}+i,T_{min(d/dt)}+10+i\right],$$
(21)$$\bar{W_{3}}<0.9,\;\;\;i=0,0.01,0.02,0.03,...,$$
(22)$$\bar{W_{4}}>0.02,\;\;\;i=0,0.01,0.02,0.03,...$$

Similarly, the start and the end of the pre-sitting turning are obtained through Equations (Equation 23) and (Equation 24), respectively:(23)$$turn\;2_{i}=min\left(W_{3}\right),$$
(24)$$turn\;2_{f}=max\left(W_{4}\right).$$

Figure 10 shows the final result of the search method and Figure 11 indicates the proposed turning activities’ identification method.

## 3. Results

The data were analyzed using MATLAB R2017b (The MathWorks, Inc., Natick, MA, USA) to obtain the results presented in this section. The video recording was analyzed using the Wondershare Filmora version 8.4.0 video editor (Wondershare Software Ltd., Shenzhen, China).

### 3.1. IMU Measurements Validation versus the Standard Clinical Procedure

To evaluate the performance of the proposed methodology in relation to the typical visual clinical procedure, the total time from each TU & Go test captured in videos for young population was tabulated obtained as the average of the times observed frame by frame by two different evaluators to compensate the bias introduced by the subjectivity present in the real practice. This was used to observe the measurements error and the Pearson correlation coefficient with respect to the typical clinical application to assess the strength of the association between the total time from each TU & Go test captured by video and the IMU measurements. The results are tabulated in Table 2 and shown in Figure 12.

The measurement system with the proposed segmentation algorithm and methodology could deliver data similar to those obtained by the observational clinical application of the TU & Go test measured in the videos for young population (see Figure 12). Here, a Pearson correlation coefficient of 0.9884 (see Figure 12a) and a mean error of 0.17 ± 0.13 s was observed, as shown in Figure 12b.

In addition, the proposed methodology in the older adults was used to evaluate the performance in the target population of this procedure. In addition, two different evaluators analyzed the videos of the tests carried out by each subject independently, the results of Table 3 and Figure 13 being obtained.

The measurement system with the proposed methodology has a similar performance to that obtained for healthy young subjects, with a Pearson correlation coefficient of 0.9878 (see Figure 13a) and a mean error of 0.20 ± 0.22 s as shown in Figure 13b. Moreover, the proposed methodology was capable of correctly classifying the 92% of the subjects measured according to their risk of falling, taking into account the total time of the test assessed by the IMU, including subjects with a high risk of falling (see subject 12 of Table 3).

### 3.2. Activities Segmentation Analysis of the TU & Go Test

To evaluate the performance of the proposed segmentation algorithm, the recording videos of the tests of each subject were analyzed and the time of each stage was tabulated to compare it with the times obtained when processing the inclination data (Pitch) and the sensor rotation data (Yaw).

Figure 14 shows the correlation for the activity segmentation time for the young subjects, showing that the minimum correlation is obtained in the identification of the transition between the end of the standing and the start of the first walk with a Pearson correlation coefficient of 0.8138 (see Figure 14a). The best correlation occurs in the identification times of the transition between the end of the 3-m turning and the start of the second walk with a Pearson correlation coefficient of 0.9854 (see Figure 14d). The above indicates that the measurement system has a high degree of agreement with respect to a visual segmentation.

Figure 15 shows the measurement errors between the proposed segmentation methodology for each sub-task during the TU & Go test in the young subjects compared to the video analysis. In Figure 15a, it is observed that the segmentation algorithm is capable of identifying the transition between the end of the standing and the start of the first walk with an average error of −0.02 s. Figure 15b shows that the transition between the end of the first walk and the start of the 3-m turning is identified with an average error of 0.36 s. Figure 15c shows that the transition between the end of the 3-m turning and the start of the second walk is identified with an average error of 0.11 s. Figure 15d shows that the transition between the end of the second walk and the start of the pre-sitting turn is identified with an average error of 0.25 s. The sub-task of the end of the pre-sitting turning is identified with an average error of 0.16 s (see Figure 15e). Regarding the sitting sub-task, the algorithm identifies the start with an average error of 0.18 s (see Figure 15f) and the end with an average error of 0.24 s (see Figure 15g).

Figure 16 shows the measurement errors between the proposed segmentation methodology for each sub-task during the TU & Go test in the older adults group compared to the video analysis. In Figure 16a, it is observed that the segmentation algorithm is capable of identifying the transition between the end of the standing and the start of the first walk with an average error of 0.07 s. Figure 16b shows that the transition between the end of the first walk and the start of the 3-m turning is identified with an average error of 0.29 s. Figure 16c shows that the transition between the end of the 3-m turning and the start of the second walk is identified with an average error of 0.43 s. Figure 16d shows that the transition between the end of the second walk and the start of the pre-sitting turn is identified with an average error of 0.63 s. The sub-task of the end of the pre-sitting turning is identified with an average error of 0.21 s (see Figure 16e). Regarding the sitting sub-task, the algorithm identifies the start with an average error of 0.25 s (see Figure 16f) and the end with an average error of 0.26 s (see Figure 16g).

### 3.3. Characterization of the Signals Acquired from the Stages of the TU & Go Test

After the segmentation, each action performed by the subjects is characterized independently, and the following variables are obtained: duration of each sub-task, standing acceleration (Acc. Su), sitting acceleration (Acc. Sd), rotation velocity of the 3-m turning (Vel. $T_{1}$), rotation velocity of the pre-sitting turning (Vel. $T_{2}$), number of steps during the first walk ($W_{1}$), number of steps during the second walk ($W_{2}$), number of steps during the 3-m turning ($T_{1}$), number of steps during the pre-sitting turning ($T_{2}$), inclination degrees of the trunk during standing (Pitch Su) and sitting (Pitch Sd) as shown in Figure 17.

The time duration of the sub-tasks performed by the young subjects are shown in Figure 18 and in Table 4. In Table 5, the parameters extracted from the signals of angular velocity and acceleration of the inertial sensors of the IMU are observed.

Figure 18a illustrates that the standing times obtained by the segmentation algorithm have a distribution close to the values observed by the video recordings, with an average error of −0.08 ± 0.15 s. Figure 18b presents one of the most different distributions in the times obtained for the first walk with an average error of 0.42 ± 0.20 s. Figure 18c presents a less dispersed distribution for the 3-m turning times than the tabulated ones from the video recording, estimated with an error of −0.19 ± 0.21 s. Regarding to the second walk times (see Figure 18d) and sitting times (see Figure 18f), an estimation close to the tabulated values are observed from the video analysis, with an error of 0.17 ± 0.11 s and −0.01 ± 0.19 s, respectively. Finally, the pre-sitting turning times are estimated with an error of −0.08 ± 0.11 s, despite the slight difference in the distributions observed in Figure 18e.

On the other hand, the time duration of the sub-tasks performed by the older adults group are shown in Figure 19 and in Table 6. In Table 7, the parameters extracted from the signals of angular velocity and acceleration of the inertial sensors of the IMU are observed.

In Figure 19, it is observed that the proposed method delivers similar distributions to those measured by video, but less variable, which is due to the present subjectivity when discriminating the transition from one sub-task to another visually. The atypical data in all the sub-images of Figure 19 are due to the subjectivity with a high risk of falling measured, demonstrating that the algorithm and the measurement methodology used allows identifying it as well as the visual method, but automatically. In general, the data presented in Table 6 indicate that the older adults group delay more than the young subjects in perform each sub-task (see Table 4), maintaining a similar error with respect to that observed in video.

From Table 5 and Table 7, it can be observed that the parameters that most differ between the two populations measured—young and older adults—are the maximum speed of both turnings, being lower in the elderly subjects than in young subjects. The largest number of steps measured for each walking sub-task of Table 7 belong to subject 12 with a high risk of falls and which is also possible to measure automatically using a simple local peak detection algorithm on the acceleration signal once the segmentation is performed.

## 4. Discussion

In this study, a segmentation of the TU & Go test activities using a single wireless IMU was performed in two different age groups. For this, a comparison between the measurements obtained by the typical observational analysis and the proposed methodology, the segmentation of the TU & Go test activities and the characterization of the inertial signals acquired from the TU & Go test stages were carried out with interesting results.

Effective treatment, specifically for gait disturbances and for risk of falls assessment, requires reliable tools. Tridimensional motion capturing systems are the gold standard of gait assessment, but, due to the space and time requirements and the high cost of the equipment, its use in clinical practice is far from routine [29]. In addition, a limited number of consecutive strides can be measured and they require camera and markers that may limit their use in the clinical practice.

In this context, IMUs may be used to assess gait performance and risk of falls. Moreover, these technologies present several advantages such as they are portable, low-cost and lightweight; they are good at measuring acceleration and turns, they are suitable for measuring brief, high speed events and they can be used indoors and outdoors regardless of lighting conditions; finally, they can continuously evaluate over long periods of time. However, they must answer to their usefulness as a clinical tool [30]. Thus, these technological devices must be assessed in terms of clinical feasibility and psychometric properties [31,32].

The proposal model obtained a high correlation respect to video recording (see Figure 12), allowing for identifying all the events with only a single wireless IMU which present a low variability in characteristics inter-subjects. It allows for extracting characteristics of each phases independently, in comparison with other works that use a body fixed sensor array [11,16]. In general, during the proof of concept of the proposed methodology in young subjects, a high degree of concordance was obtained with respect to the segmentation performed by video, which is demonstrated in the correlation coefficients of Figure 14. In the case of the group of older adults, the correlation analysis was omitted because the results could be unrepresentative due to the low number of subjects and the atypical data present in the subject with a high risk of falls (see subject 12 of Table 3). The worst correlation obtained was found in the identification of the moment in which the person finishes the sit-to-stand transition, due to the difficulty to identify in a visual way the exact point of the end of the standing and the beginning of the first walk sub-task. However, the proposed analysis algorithm was capable of identifying the exact transition point with an average maximum error of 0.36 s (see Figure 15b) for young subjects and a maximum average error of 0.63 s for older adults (see Figure 16d).

The segmentation presented in the present study allows for improving objectivity to the clinical practice in the evaluation of the performance of the patients. By observing the standing/sitting phases using Pitch, it is possible to determine the inclination of the trunk during sit to walk transfer in young subjects (see Table 5) and older adults subjects (see Table 7. This is relevant because, as mentioned by Pozaic et al., the transition from sit to walk is the event with the highest number of falls in the elderly population [33].

On the other hand, the analysis of the turning phases through Yaw will allow for observing the degrees and the time that the subject takes to carry out this activity (Figure 18c,e). The ability to turn safely is relevant to functional independence and is considered another difficult task with a high risk of falling [34,35].

Although the conventional TU & Go test is a clinical tool that allows for measuring the risk of falling, it uses a single global parameter to estimate it. However, the stage that could be interfering in the execution of the task in less time is not specifically discriminate. The current methodology provides information about variables such as: standing acceleration (Acc. Su), sitting acceleration (Acc. Sd), rotation velocity of the 3-m turning (Vel. $T_{1}$), rotation velocity of the pre-sitting turning (Vel. $T_{1}$), number of steps during the first walk (Steps $W_{1}$), number of steps during the second walk (Steps $W_{2}$), number of steps during the 3-m turning (Steps $T_{1}$), number of steps during the pre-sitting turning (Steps $T_{2}$), inclination degrees of the trunk during standing (Pitch Su) and sitting (Pitch Sd) (see Table 5).

Other authors [36,37,38,39] have explored the use of IMU technology for gait assessment during the TU & Go test. However, this approach allows for segmenting the transitions between each sub-task in an exact and automatic way, using a simple algorithm and a low–cost movement sensor, allowing for extracting characteristics of each one of them due to the positioning of the IMU in the young and older adults groups. The above can be extrapolated when performing tests of 6 and 10 m to extract more parameters of the gait during the first and second walk.

This study presents some limitations. Firstly, the segmentation of each event during the analysis of the videos was performed in an observational way. Second, it is not possible to determine how the algorithm would perform in other populations such as people with neurological disorders. However, the whole sample recruited was determined as the starting point to be capable of validating the algorithm and the sensors used in pathological population in future research. Clinical validation studies of these devices should be carried out in populations with specific characteristics related to gait and balance impairments.

## 5. Conclusions

In conclusion, an IMU located on the back may detect main gait events and spatial-temporal kinematic factors during the TU & Go test, with excellent correlation with the conventional visual clinical procedure in young and older adults. Thus, a user-friendly technological tool to health-care professionals may offer objective measurements for the segmentation of the activities as standing, walking, turning and sitting related to critical events related with the risk of falls.

## Figures and Tables

**Figure 1 sensors-19-01647-f001:**
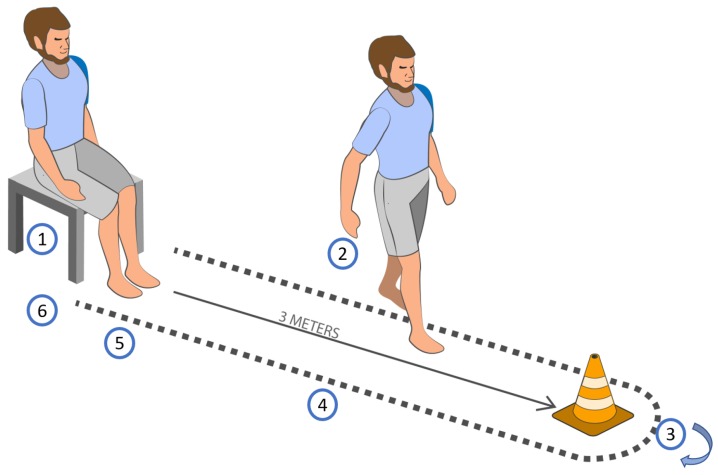
Activities performed in the 3-m TU & Go test. 1 = standing, 2 = first walk, 3 = 3-m turning, 4 = second walk, 5 = pre-sitting turning, and 6 = sitting.

**Figure 2 sensors-19-01647-f002:**
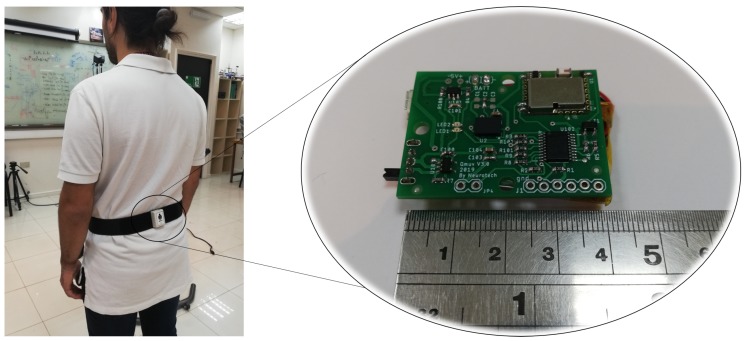
IMU sensor develop in the biomedical engineering laboratory of the Universidad de Concepción, Chile.

**Figure 3 sensors-19-01647-f003:**
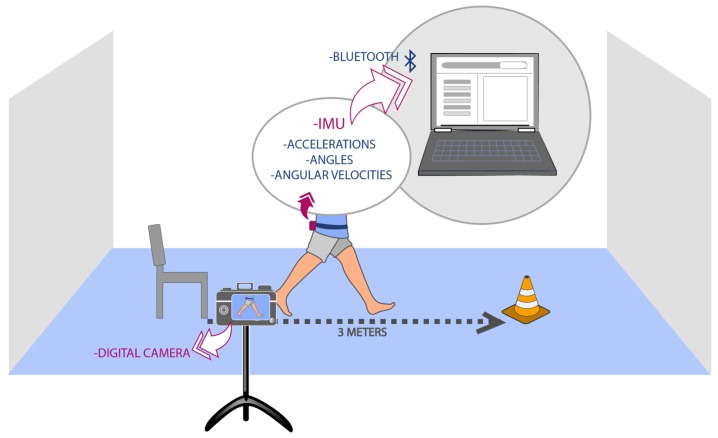
Set up used for the acquisition data during 3-m TU & Go test.

**Figure 4 sensors-19-01647-f004:**
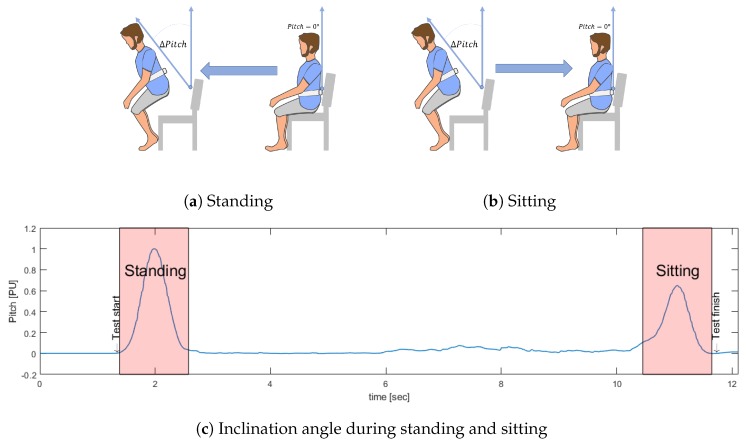
Identification of Standing/Sitting events using Pitch.

**Figure 5 sensors-19-01647-f005:**
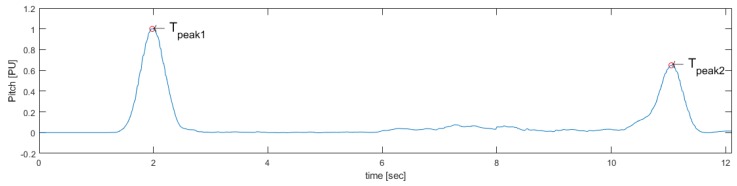
Peaks identified for the search of start and end of standing and sitting actions.

**Figure 6 sensors-19-01647-f006:**
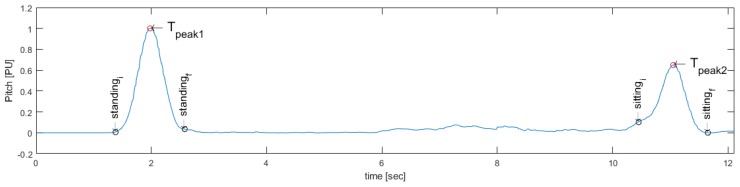
Result of the proposed method for the search of the beginning ($standing_{i}$) and the end ($standing_{f}$) of the Standing event and for the search of the beginning ($sitting_{i}$) and the end ($sitting_{f}$) of the Sitting event.

**Figure 7 sensors-19-01647-f007:**
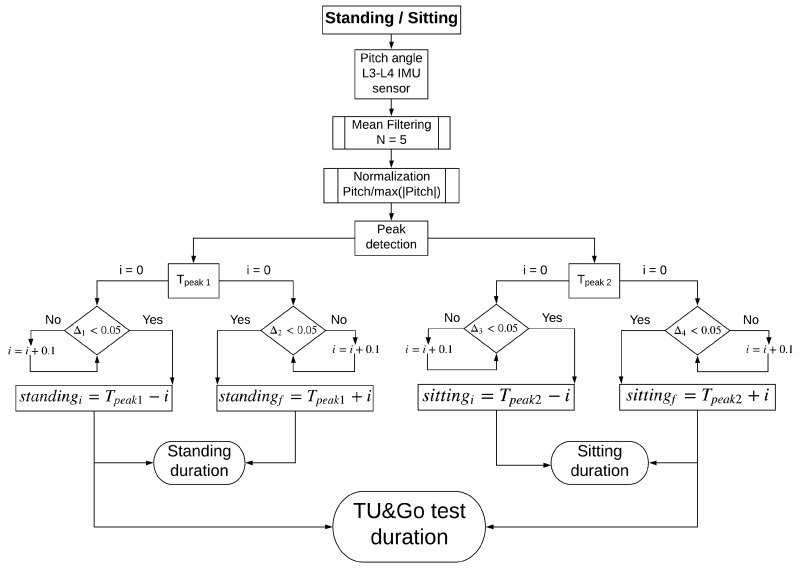
Automatic feature-based segmentation algorithm for the standing and sitting activities in the 3–m TU & Go test.

**Figure 8 sensors-19-01647-f008:**
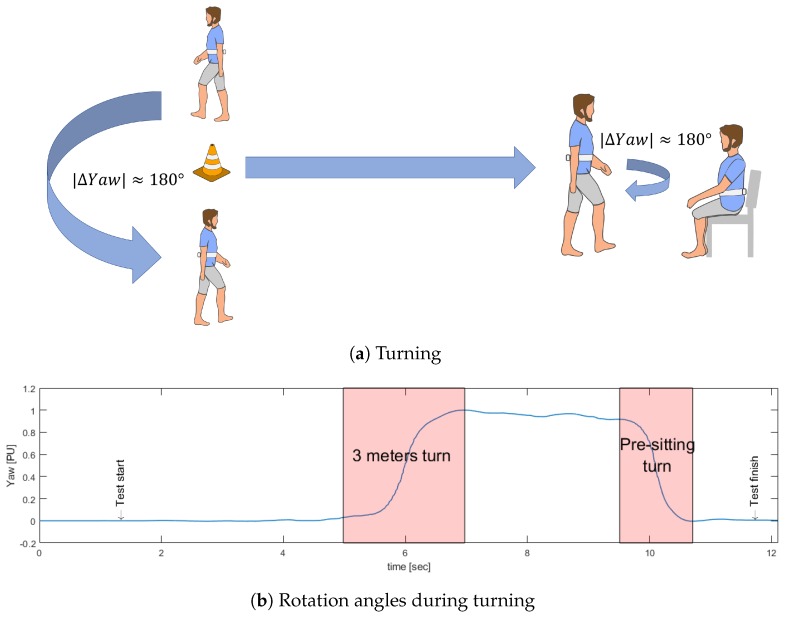
Turning events identification using Yaw.

**Figure 9 sensors-19-01647-f009:**
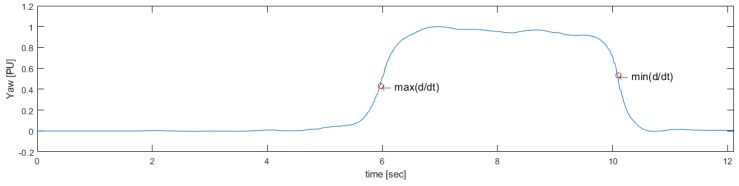
Maximum and minimum of the rotation signal derived to start the search for turning events.

**Figure 10 sensors-19-01647-f010:**
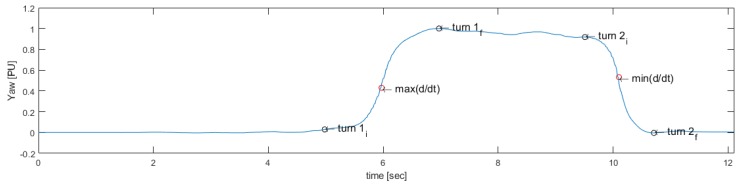
Result of the proposed method for the search of the beginning ($turn1_{i}$) and the end ($turn1_{f}$) of the 3-m turning event and for the search of the beginning ($turn2_{i}$) and the end ($turn2_{f}$) of the pre-sitting event.

**Figure 11 sensors-19-01647-f011:**
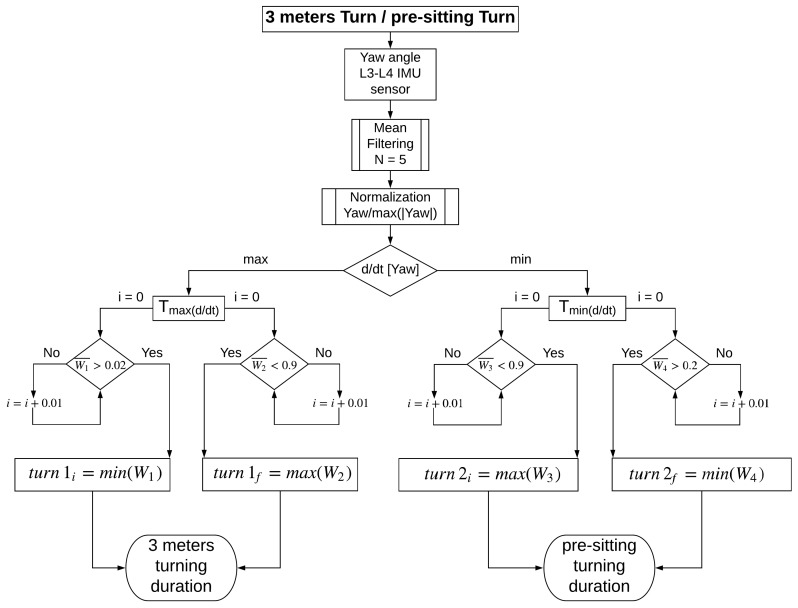
Automatic feature-based segmentation algorithm for the turning actions in the 3-m TU & Go test.

**Figure 12 sensors-19-01647-f012:**
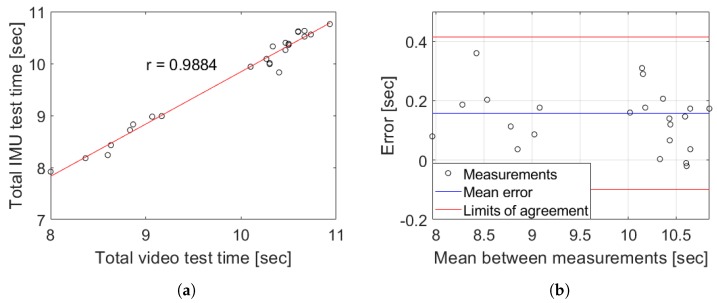
Results from the measurements obtained by the proposed methodology versus the typical visual clinical procedure in young subjects. (**a**) Pearson correlation between IMU and the video record analysis; (**b**) Bland–Altman plot between IMU and the video record analysis.

**Figure 13 sensors-19-01647-f013:**
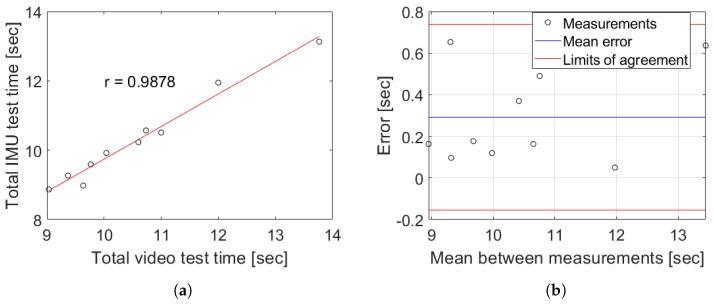
Results from the measurements obtained by the proposed methodology versus the typical visual clinical procedure in the older adults group. (**a**) Pearson correlation between IMU and the video record analysis without the high risk of falling subject; (**b**) Bland–Altman plot between IMU and the video record analysis.

**Figure 14 sensors-19-01647-f014:**
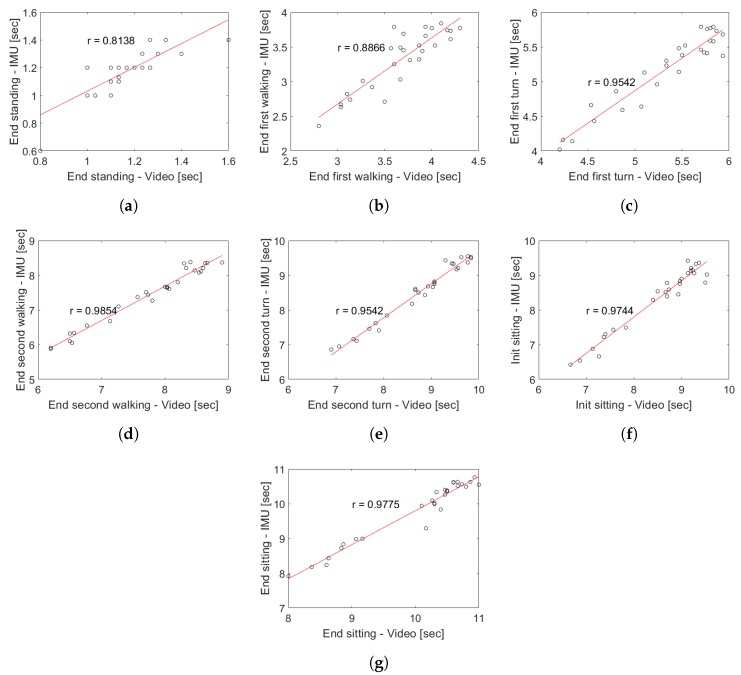
Pearson correlation coefficient for the segmentation time for each sub task of the TU & Go test in young subjects. (**a**) end of the standing/start of the first walk; (**b**) end of the first walk/start of the 3-m turn; (**c**) end of the 3-m turn/start of the second walk; (**d**) end of the second walk/start of the pre-sitting turn; (**e**) end of the pre-sitting turn; (**f**) start of the sitting; (**g**) end of the sitting.

**Figure 15 sensors-19-01647-f015:**
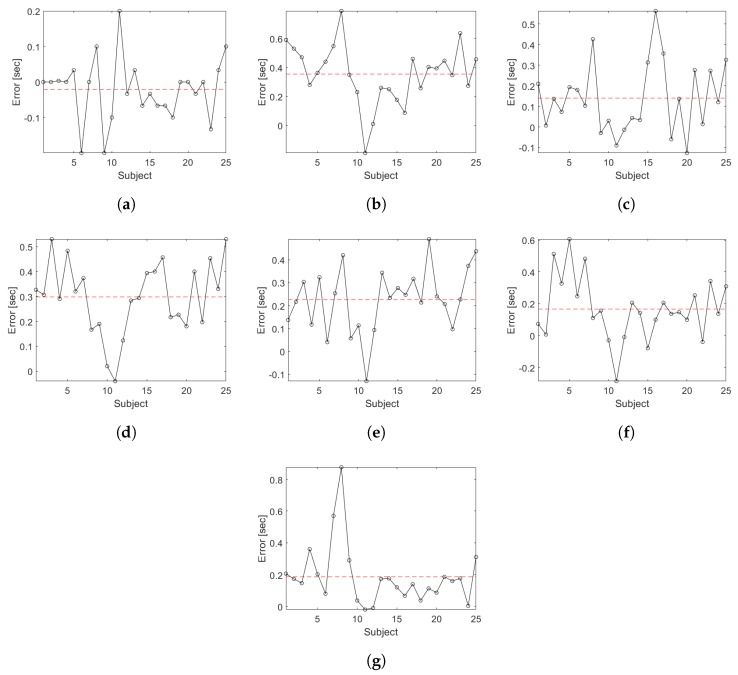
Error plots for the segmentation time for each sub task of the TU & Go test in young subjects. The segmented lines correspond to the mean error. (**a**) end of the standing/start of the first walk; (**b**) end of the first walk/start of the 3-m turn; (**c**) end of the 3-m turn/start of the second walk; (**d**) end of the second walk/start of the pre-sitting turn; (**e**) end of the pre-sitting turn; (**f**) start of the sitting; (**g**) end of the sitting.

**Figure 16 sensors-19-01647-f016:**
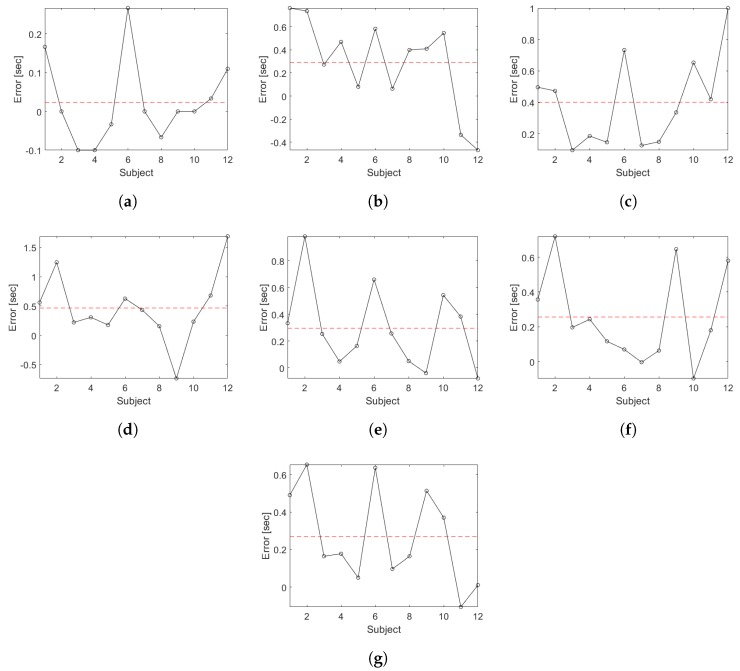
Error plots for the segmentation time for each sub task of the TU & Go test in the older adults group. The segmented lines correspond to the mean error. (**a**) end of the standing/start of the first walk; (**b**) end of the first walk/start of the 3-m turn; (**c**) end of the 3-m turn/start of the second walk; (**d**) end of the second walk/start of the pre-sitting turn; (**e**) end of the pre-sitting turn; (**f**) start of the sitting; (**g**) end of the sitting.

**Figure 17 sensors-19-01647-f017:**
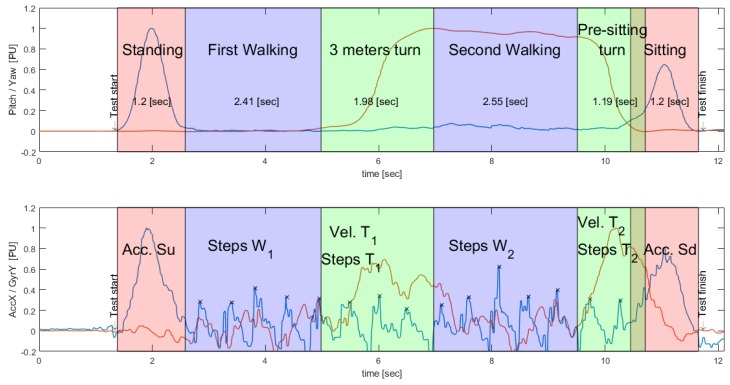
Characteristics measured after the segmentation of the activities carried out in the 3-m TU & Go test. The measured characteristics correspond to the signals of vertical acceleration (AccZ) and sagittal angular velocity (GyrY).

**Figure 18 sensors-19-01647-f018:**
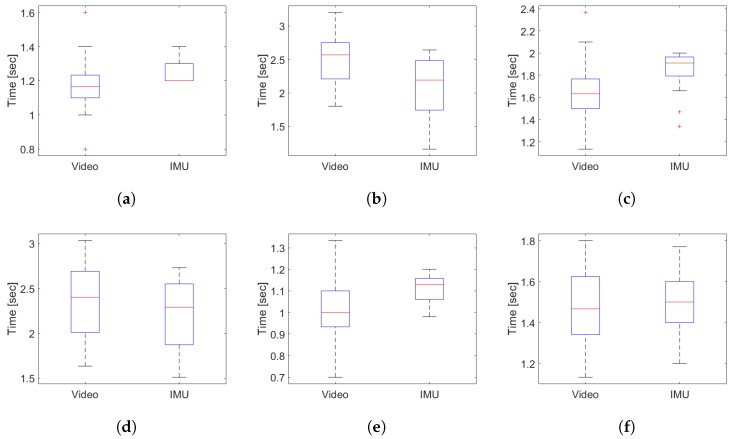
Distribution plots of the duration times obtained with the segmentation algorithm compared with the video analysis in the young subjects. (**a**) duration distribution for standing time; (**b**) duration distribution for the first walk time; (**c**) duration distribution for the 3-m turning; (**d**) duration distribution for the second walk time; (**e**) duration distribution for the pre-sitting turning time; (**f**) duration distribution for sitting time.

**Figure 19 sensors-19-01647-f019:**
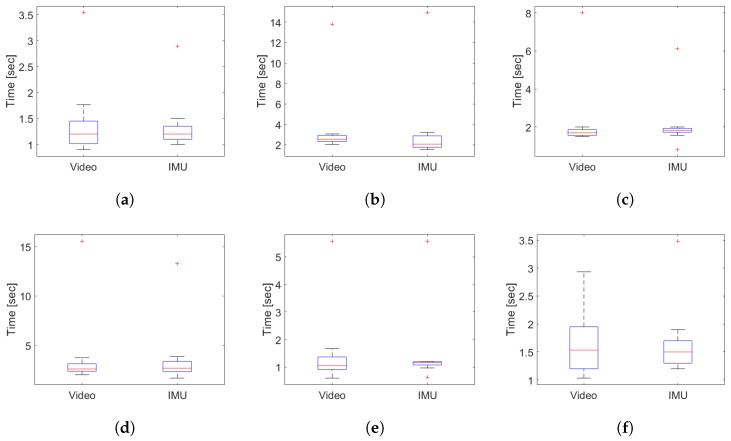
Distribution plots of the duration times obtained with the segmentation algorithm compared with the video analysis in the older adults group. (**a**) duration distribution for standing time; (**b**) duration distribution for the first walk time; (**c**) duration distribution for the 3-m turning; (**d**) duration distribution for the second walk time; (**e**) duration distribution for the pre-sitting turning time; (**f**) duration distribution for sitting time.

**Table 1 sensors-19-01647-t001:** Characteristics of the inertial and magnetic sensors of the IMU used.

Sensor	Axis	Range	Bandwidth	Resolution	Output Rate
Accelerometer	XYZ	±16 G	62.5 Hz	14 bits (≈1.95 mG)	100 Hz
Gyroscope	XYZ	±2000 dps	32 Hz	16 bits (≈0.061 dps)	100 Hz
Magnetometer	XY	±1300 [μT]	10 Hz	13 bits (≈317 [ηT])	20 Hz
-	Z	±2500 [μT]	10 Hz	15 bits (≈152 [ηT])	20 Hz

**Table 2 sensors-19-01647-t002:** Results from the measurements obtained by the proposed methodology versus the typical visual clinical procedure in young subjects.

Subject	Height [cm]	Weight [kg]	Age [years]	Gender	Total Video Time s	Total IMU Time s
Subject 1	179	93	25	male	10.46	10.26
Subject 2	168	70	25	male	10.73	10.56
Subject 3	176	81	28	male	10.66	10.52
Subject 4	164	68	26	female	8.60	8.24
Subject 5	160	68	25	female	8.63	8.43
Subject 6	162	70	29	male	8.00	7.92
Subject 7	168	80	26	female	10.40	9.83
Subject 8	152	48	25	female	10.16	9.29
Subject 9	195	90	26	female	10.30	10.01
Subject 10	190	80	29	male	10.66	10.63
Subject 11	169	68	29	male	10.60	10.62
Subject 12	153	77	27	male	10.60	10.61
Subject 13	160	73	33	female	10.93	10.76
Subject 14	172	81	31	female	10.26	10.09
Subject 15	168	70	26	male	10.50	10.38
Subject 16	173	79	28	male	10.46	10.40
Subject 17	170	73	25	male	10.50	10.36
Subject 18	163	60	32	male	8.86	8.83
Subject 19	175	70	28	male	8.83	8.72
Subject 20	171	68	33	male	9.06	8.98
Subject 21	165	70	32	male	8.36	8.18
Subject 22	179	73	26	male	10.10	9.94
Subject 23	185	79	26	male	9.16	8.99
Subject 24	190	83	30	male	10.33	10.33
Subject 25	160	58	29	male	10.30	9.99

**Table 3 sensors-19-01647-t003:** Results from the measurements obtained by the proposed methodology versus the typical visual clinical procedure in the older adults group, where rof = risk of falling, high = high risk of falling, low = low risk of falling and no = no risk of falling. The classification of the risk of falling of the subjects was obtained by following the manual of the ministry of health (MINSAL) for the Chilean population [28].

Subject	Height [cm]	Weight [kg]	Age [years]	Gender	Total Video Time s (rof)	Total IMU Time s (rof)
Subject 1	168	75	60	male	11.00 (low)	10.51 (low)
Subject 2	168	72	60	male	9.63 (no)	8.98 (no)
Subject 3	156	90	63	male	10.73 (low)	10.57 (low)
Subject 4	170	68	65	male	9.76 (no)	9.59 (no)
Subject 5	179	64	71	male	12.00 (low)	11.95 (low)
Subject 6	157	62	60	female	13.76 (low)	13.13 (low)
Subject 7	178	93	67	male	9.36 (no)	9.27 (no)
Subject 8	160	51	63	female	9.03 (no)	8.87 (no)
Subject 9	145	70	59	female	10.43 (low)	9.92 (no)
Subject 10	173	84	59	male	10.60 (low)	10.23 (low)
Subject 11	160	68	59	female	9.33 (low)	9.44 (low)
Subject 12	157	61	93	female	43 (high)	42.99 (high)

**Table 4 sensors-19-01647-t004:** Average time duration and error from the IMU activities segmentation algorithm compared to the video analysis of the young subjects.

	Standing	First Walk	3-m Turning	Second Walk	Pre-Sitting Turning	Sitting
Video duration s	1.16 ± 0.15	2.51 ± 0.35	1.65 ± 0.25	2.39 ± 0.36	1.03 ± 0.14	1.48 ± 0.17
IMU duration s	1.24 ± 0.07	2.10 ± 0.45	1.85 ± 0.16	2.22 ± 0.37	1.10 ± 0.06	1.48 ± 0.15
IMU error estimation s	−0.08 ± 0.15	0.42 ± 0.20	−0.19 ± 0.21	0.17 ± 0.11	−0.08 ± 0.11	−0.01 ± 0.19

**Table 5 sensors-19-01647-t005:** Characteristics obtained after the implemented segmentation of the activities performed by the young subjects during the TU & Go test.

	Acc. Su [m/s${}^{2}$]	Acc. Sd [m/s${}^{2}$]	Vel. $T_{1}$ [deg/s]	Vel. $T_{2}$ [deg/s]	Steps $W_{1}$	Steps $W_{2}$	Steps $T_{1}$	Steps $T_{2}$	Pitch Su [deg]	Pitch Sd [deg]
Minimum	3.9102	4.459	126.3125	181.6875	2	3	2	1	2.2979	3.236
Maximum	9.1532	7.497	240.6875	264.5625	7	6	5	3	8.9954	9.9904
Mean	6.49 ± 1.42	6.09 ± 0.87	156.25 ± 28.49	213.62 ± 22.31	4.29 ± 1.29	4.22 ± 1.05	3.55 ± 0.89	1.88 ± 0.80	5.35 ± 1.89	6.25 ± 1.82

**Table 6 sensors-19-01647-t006:** Average time duration and error from the IMU activities segmentation algorithm compared to the video analysis of the older adults group.

	Standing	First Walk	3-m Turning	Second Walk	Pre-Sitting Turning	Sitting
Video duration s	1.41 ± 0.71	3.49 ± 3.26	2.23 ± 1.82	3.75 ± 3.75	1.32 ± 0.14	1.65 ± 0.56
IMU duration s	1.36 ± 0.50	3.25 ± 3.70	2.08 ± 1.30	3.56 ± 3.13	1.46 ± 1.30	1.64 ± 0.62
IMU error estimation s	0.05 ± 0.30	0.23 ± 0.61	0.14 ± 0.62	0.19 ± 0.78	−0.01 ± 0.27	0.01 ± 0.29

**Table 7 sensors-19-01647-t007:** Characteristics obtained after the implemented segmentation of the activities performed by the older adults group during the TU & Go test.

	Acc. Su [m/s${}^{2}$]	Acc. Sd [m/s${}^{2}$]	Vel. $T_{1}$ [deg/s]	Vel. $T_{2}$ [deg/s]	Steps $W_{1}$	Steps $W_{2}$	Steps $T_{1}$	Steps $T_{2}$	Pitch Su [deg]	Pitch Sd [deg]
Minimum	3.89	3.15	43.87	62.06	3	3	3	1	2.82	1.33
Maximum	9.52	10.16	228.18	295.43	35	32	10	7	11.06	15.71
Mean	7.20 ± 1.63	7.08 ± 2.18	138.75 ± 42.21	189.79 ± 57.38	6.91 ± 8.89	7.66 ± 7.81	4.62 ± 1.92	2.58 ± 1.62	7.48 ± 2.81	8.68 ± 4.59

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
