# Peer review of "Use of a Single Wireless IMU for the Segmentation and Automatic Analysis of Activities Performed in the 3-m Timed Up & Go Test"

_sensors, 2019, doi:10.3390/s19071647_

Reviewer 1 Report

The work presents a segmentation methodology based on a single wireless IMU in order to analyse the data of the inertial sensors, during a TU&Go test, to provide a complete report on risks of falls. A set of experiments is presented to support the evaluation of the methodology and to be compare against the gold standard, but they have serious flaws: only young healthy persons are enrolled; a chair without armrests is used, so old persons probably are not able to stand up at all; the used IMU is homemade, why; the considered events are very simple, at least for the persons performing the test.

A lot of papers on using IMU during a TU&Go test and aiming at the test phases individuation exists in the literature, for example:

https://doi.org/10.3390/s17040934

https://doi.org/10.1109/EMBC.2015.7319556

https://doi.org/10.1109/MOCAST.2018.8376592

https://doi.org/10.1016/j.gaitpost.2012.02.006

https://doi.org/10.1016/j.parkreldis.2010.08.001

Author Response

Dear Reviewer,

We appreciate the comments on suggestions sent. We detail below the modifications that have been made, 

Considering the suggestion given about studied population, we decided to evaluate 12 older adults between 59 and 93 years old that fulfilled the inclusion and exclusion criteria. This data are presented in the Results, section 3.1, table 3 and figure 4; in section 3.2 figure 16; in section 3.3, figure 19 and tables 6 and 7. 

About the use of a chair without armrests, we used the procedure proposed by Salarian et al. in reference [22], due to both groups measured are healthy subjects, who did not need armrests to perform the standing and sitting.

In relation to the homemade IMU, according to the given suggestion, a detail description of the device was added in section 2.2 of Materials and Methods. 

For the segmentation, the characteristics events of the TUG were selected, which are analyzed mainly in clinical practice and in other related studies, as mentioned in the introduction and discussion sections. Undoubtedly, more complex events of the subject´s gait could be evaluated, modifying the test to 6 o 10 meters, which is not the main focus of this study. 

We reviewed the suggested studies and they were added in the discussion section, between lines 326 and 331.

Finally, the translation of the manuscript was completely reviewed by native translator. 

Again, we appreciate the suggestions and hope that all modifications can improve the quality and understanding of our manuscript, in order to comply whit the standard of the journal.

Sincerely,

Pablo Aqueveque Navarro, PhD.

Corresponding Author. 

Reviewer 2 Report

The authors present a study that uses a single wireless IMU to segment the standard Timed Up and Go (TUG) task. While this approach has potential importance within the balance and falls community, there are several critical limitations.

Major comments:

The authors describe and validate their algorithm in healthy younger adults, when the target population is older adults at risk of falling. While generally acceptable as a proof of concept, the use of younger adults is problematic in this case as several firm thresholds are proposed, which may not hold true for older adults (e.g., patients with Parkinson's disease may take longer to turn than the maximum 1.2 seconds allowed in the algorithm). While it is mentioned the subjects were videotaped, there is no mention of how these videos were coded or analyzed. Were two independent raters used?

The authors use an in-house device that estimates sensor orientation. A brief, at a minimum, description of the sensor fusion algorithm they use to estimate orientation is required.

Additionally, specific comparisons between the IMU and the video analysis are reported for total time only, yet the focus of the manuscript is on the segmentation. In Table 1, it appears the error in some segmentations is between 10-20%. Bland-Altman plots, rather than boxplots in Figure 12, are necessary to compare the bias and error between systems.

Author Response

Dear Reviewer,

We appreciate the comments and suggestions sent. We detail below the modifications that have been made,

Considering the suggestion given about the studied population, we decided to evaluate 12 older adults between 59 and 93 years old that fulfilled the inclusion and exclusion criteria. This data are presented in the Results, section 3.1, table 3 and figure 4; section 3.2, figure 16; section 3.3, figure 19 and tables 6 and 7. 

Considering that the time in older adults in the execution of turning may be bigger than the window of analysis applied to young subjects, the algorithm was modified, which is shown in Material and Methods, section 2.4.2.

For an improvement in the understanding of the video analysis procedure, it is detailed in the Results, section 3.1 between lines 180 y 183. 

In relation to the homemade IMU, according to the given suggestion, a detailed description of the device was added in section 2.2 of Materials and Methods. 

Modifications were made in the analysis comparing the segmentation of the IMU vs Video, adding it in Results, section 3.2.

Finally, the translation of the manuscript was completely reviewed by a native translator. 

Again, we appreciate the suggestions and hope that our modifications can improve the quality and understanding of our manuscript, in order to comply whit the standard of the journal.

Sincerely, 

Pablo Aqueveque Navarro, PhD.

Corresponding Author. 

Reviewer 3 Report

A single wireless IMU placed on the waist is used to segment tasks in the standard Timed Up and Go test. Although the idea has potential in order to assess balance and fall risk in the aging people, critical limitations and research design inaccuracies need to be addressed.

Major comments:

One of the major critical point is the validation of the proposed algorithm with very young and healthy subjects. This may be misleading and it might not be transferred to an older population, subject to fall risk, expecially becouse some interval and threshold are a priori imposed. 

Very few information are given for the homemade IMU (accuracy, precision, range of measurements, etc…), was it tested and compared to a golden standard measurement system? Please insert references. The same concerns about the internal processor, no information are provided.

Authors claim that a GOPRO camera was used. Although this can be used to check consistence of IMU results, it cannot be referred as golden standard, also due to misalignment errors that I expect to be present. Moreover no information how video were post-processed, how data were extracted and how many independent observers were involved 

Some punctual observations:

L. 126: reference for the sentence “which can even be observed in subjects with reduced mobility”

L. 135 and following. It seems to me that there is rather a confusion between the value of the peak Tpeak (figure 4) and the time/sample  at which the peak occurs. I guess i refers to the sample number, but then it in the formula Tpeak and i are added.

L 139 Figure 6 shows the proposed

Results:  in some cases the differences of time are lower than the acquisition frequency. This results cannot be reported. Also some SD are reported with too many decimal number, which are not significative.

Figure 13: in the middle of the lower graph there is T2, I believe it is T1

Table 2: max and min values are exchanged.

English has to be checked and changes in some sentences are needed.

Author Response

Dear Reviewer,

We appreciate the comments and suggestions sent. We detail below the modifications that have been made, 

Considering the suggestion given about the studied population, we decided to evaluate 12 older adults between 59 and 93 years old that fulfilled the inclusion and exclusion criteria. This data are presented in the Results, section 3.1, table 3 and figure 4; section 3.2, figure 16; section 3.3, figure 19 and tables 6 and 7.

In relation to the homemade IMU, according to the given suggestion, a detailed description of the device was added in section 2.2 of Materials and Methods. 

Regarding the use of the concept "Gold-Standard" to refer to the video analysis, it was modified by "typical clinical application".

For an improvement in understanding of the video analysis procedure it is detail in the Results, section 3.1 between lines 180 and 183.

All the issues and observations indicated were included.

Finally, the translation of the manuscript was completely reviewed by a native translator.

Again, we appreciate the suggestions and hope that our modifications can improve the quality and understanding of our manuscript, in order to comply with the standard of the journal. 

Sincerely,

Pablo Aqueveque Navarro. PhD.

Corresponding Author.

Round  2

Reviewer 1 Report

The authors improved the manuscript and answered to all the raised questions. Even the english is better.

Reviewer 2 Report

The authors have made modifications to the manuscript to address all previous comments

Reviewer 3 Report

I appreciate the efforts authors made to amend the paper according to reviewers’ suggestions.

English was also improved.  The paper can be accepted for publication.